# Targeted degradation of zDHHC-PATs decreases substrate S-palmitoylation

**Mingjie Bai[1], Emily Gallen[1], Sarah Memarzadeh[2], Jacqueline Howie[1], Xing Gao[1], Chien-Wen S. Kuo[1], Elaine Brown[1], Simon Swingler[3], Sam J. Wilson[3], Michael J. Shattock[4], David J. France[2]*, William Fuller[1]***

1 School of Cardiovascular & Metabolic Health, College of Medical Veterinary and Life Sciences, University of Glasgow, Glasgow, United Kingdom, 2 School of Chemistry, University of Glasgow, Glasgow, United Kingdom, 3 Medical Research Council–University of Glasgow Centre for Virus Research, College of Medical Veterinary and Life Sciences, University of Glasgow, Glasgow, United Kingdom, 4 School of Cardiovascular and Metabolic Medicine & Sciences, King's College London, London, United Kingdom

* David.France@glasgow.ac.uk (DJF); Will.Fuller@glasgow.ac.uk (WF)

## Abstract

Reversible S-palmitoylation of protein cysteines, catalysed by a family of integral membrane zDHHC-motif containing palmitoyl acyl transferases (zDHHC-PATs), controls the localisation, activity, and interactions of numerous integral and peripheral membrane proteins. There are compelling reasons to want to inhibit the activity of individual zDHHC-PATs in both the laboratory and the clinic, but the specificity of existing tools is poor. Given the extensive conservation of the zDHHC-PAT active site, development of isoform-specific competitive inhibitors is highly challenging. We therefore hypothesised that proteolysis-targeting chimaeras (PROTACs) may offer greater specificity to target this class of enzymes. In proof-of-principle experiments we engineered cell lines expressing tetracycline-inducible Halo-tagged zDHHC5 or zDHHC20, and evaluated the impact of Halo-PROTACs on zDHHC-PAT expression and substrate palmitoylation. In HEK-derived FT-293 cells, Halo-zDHHC5 degradation significantly decreased palmitoylation of its substrate phospholemman, and Halo-zDHHC20 degradation significantly diminished palmitoylation of its substrate IFITM3, but not of the SARS-CoV-2 spike protein. In contrast, in a second kidney derived cell line, Vero E6, Halo-zDHHC20 degradation did not alter palmitoylation of either IFITM3 or SARS-CoV-2 spike. We conclude from these experiments that PROTAC-mediated targeting of zDHHC-PATs to decrease substrate palmitoylation is feasible. However, given the well-established degeneracy in the zDHHC-PAT family, in some settings the activity of non-targeted zDHHC-PATs may substitute and preserve substrate palmitoylation.

## Introduction

*S*-palmitoylation (also referred to as *S*-acylation), a dynamic type of lipidation, reversibly anchors proteins to membranes through attachment of a saturated fatty acid (typically palmitate) to cysteine residues *via* the formation of a thioester bond. This modification is catalysed by zDHHC-domain containing palmitoyl acyl transferase enzymes (zDHHC-PATs), and

**Data Availability Statement:** All relevant data are within the manuscript and its Supporting Information files.

**Funding:** We acknowledge support from the following funders: British Heart Foundation, RG/17/

15/33106 Professor William Fuller Medical Research Scotland, Daphne Jackson Fellowship Simon Swingler

**Competing interests:** The authors have declared that no competing interests exist.

reversed by thioesterases [1]. The zDHHC-PATs are integral membrane proteins located throughout the secretory pathway, whereas all thioesterases identified to date are cytosolic serine hydrolases.

The zDHHC-PAT family comprises 23 members in humans [2]. Some of these enzymes display overlapping substrate specificities and subcellular locations, while others are restricted to a single cellular compartment [1]. In recent years, these enzymes have emerged as potential therapeutic targets for various diseases, including cancer, neurological disorders, cardiovascular diseases, and infectious diseases [3]. For example, envelope glycoproteins from a diverse range of human viral pathogens are *S*-palmitoylated, including SARS-CoV-2, SARS-CoV, influenza A, measles, rabies, ebola and HIV-1 [4]. Other viral proteins are also known to be *S*-palmitoylated, with important consequences for virus assembly, replication, and virulence [5–10]. Functionally, *S*-palmitoylation can control viral budding and release (influenza HA [11], togavirus E1/E2 [12, 13]) as well as viral membrane fusion (influenza HA [14–16], SARS-CoV spike [17, 18], SARS-CoV-2 spike [19–21]). No known viruses encode zDHHC-PATs, implying that *S*-palmitoylation of viral proteins is entirely dependent upon host enzymes.

In cardiac muscle, the ubiquitous Na/K ATPase is vital for ion homeostasis as well as contractile and mitochondrial function [22]. In cardiac pathologies, reduced Na/K ATPase activity and elevated intracellular sodium concentrations degrade trans-sarcolemmal ion gradients, impair systolic and diastolic function, and reduce mitochondrial ATP production [23]. The accessory protein phospholemman (PLM) activates Na/K ATPase when phosphorylated [24, 25], inhibits the enzyme when palmitoylated [26, 27], and in recent years has emerged as a drug target to correct ion transport defects associated with heart failure [28].

Hence zDHHC-PATs are attractive drug targets: an agent targeting zDHHC20 (responsible for SARS-CoV-2 spike protein *S*-palmitoylation [19]) would offer significant therapeutic potential to manage pathology caused by coronaviruses and possibly other infectious agents. Inhibiting zDHHC5 palmitoylation of PLM would restore ion gradients, improving contractile function and mitochondrial ATP production in heart failure. However, there are no reliable zDHHC-PAT inhibitors in routine experimental use, let alone compounds that can selectively inhibit one zDHHC isoform over another. The widely used irreversible inhibitor 2-bromopalmitate (2-BP) has numerous off-target effects, including the thioesterase enzymes [29]. A recently described acrylamide-based inhibitor is clearly an improvement on 2-BP, but inhibits multiple zDHHC-PATs [30]. The conservation of all zDHHC-PAT active sites makes it unlikely that inhibitors targeting the active site will show isoform selectivity throughout this family (although inhibitors of zDHHC2 identified by high throughput screening show some selectivity for zDHHC2 and zDHHC15 over zDHHC3 and zDHHC7 [31]). Strategies aimed at blocking substrate recruitment offer more promise, as the intracellular N and C termini of the zDHHC-PATs are poorly conserved [32, 33]. However, the zDHHC-PAT family may also be tractable to targeted protein degradation using an emerging paradigm in medicinal chemistry known as proteolysis targeting chimeras (PROTACs) [34, 35]. A PROTAC contains both a ligand that is selectively targeted to a protein of interest and a ligand for cellular ubiquitination machinery. Once the PROTAC has bound its target, it recruits a ubiquitin ligase to the complex which results in ubiquitination of the target and subsequent degradation by the proteasome. Since PROTACs do not need to engage an enzyme's active site for efficacy, the entire intracellular region of an enzyme is a suitable target. This significantly expands the druggable surface area of a protein and offers the potential for protein families with highly homologous active sites to be selectively targeted.

The HaloTag protein is a bacterial dehalogenase that has been modified to bond covalently to a 6-carbon chloroalkane with high specificity. Attaching the chloroalkane ligand for Halo-Tag to an E3 ligase ligand generates Halo-PROTACs. These reagents have been well validated,

display no toxicity when applied to cells at concentrations below 10μM, and offer striking specificity at a whole proteome level [36, 37]. In this investigation, we set out to probe the tractability of the zDHHC-PAT family to PROTAC-mediated degradation. We employed the HaloTag system in combination with Halo-PROTACs to determine whether individual Halo-zDHHC-PATs were amenable to PROTAC mediated degradation, and whether zDHHC-PAT degradation impaired palmitoylation of individual substrates.

## Results

### Subcellular localisation of Halo-zDHHC5 and Halo-zDHHC20 in engineered Flp-In 293 T-REx cells

We used the Flp-In system to engineer two cell lines expressing tetracycline inducible Halo-zDHHC-PATs. For experiments targeting expression of and substrate palmitoylation by zDHHC5, a single expression cassette encoding both Halo-zDHHC5 and its substrate PLM separated by an internal ribosome entry site was introduced into cells. We have previously validated the Flp-In system to evaluate palmitoylation of PLM by zDHHC5 [27]. For experiments investigating zDHHC20, an expression cassette encoding only Halo-zDHHC20 was introduced. We first evaluated the subcellular location of Halo-zDHHC5 and Halo-zDHHC20 by staining with TAMRA-chloroalkane which labels Halo-tagged proteins in cells (Fig 1). No staining was detected in cells not treated with tetracycline to induce target gene expression. Halo-zDHHC5 was predominantly detected at the cell surface membrane, consistent with its established role as a cell surface zDHHC-PAT (Fig 1A). Halo-zDHHC20 was detected in both intracellular compartments and the cell surface membrane (Fig 1B). Western blotting for the Halo-tag detected both proteins migrating at the predicted sizes only in cells treated with tetracycline (Fig 1C).

### Halo-PROTAC induced degradation of Halo-zDHHC5 and Halo-zDHHC20 in Flp-In 293 T-REx cells

We based the design for our Halo-PROTACs on the cereblon ligand pomalidomide (**2**, Fig 2A) [38] classical VHL ligand VH032 (**3**), as well as a modified VHL-targeting scaffold that includes a fluorocyclopropane (**1**) [36, 37]. The inclusion of the fluorocyclopropane has been shown to lead to improve VHL binding affinity *in vitro*, with a corresponding improvement in cellular potency [37, 39]. We evaluated the ability of these different Halo-PROTACs (Fig 2A) to induce degradation of Halo-zDHHC-PATs.

Halo-zDHHC5 was successfully degraded using VHL-directed Halo-PROTACs **1** and **3** but not by cereblon-directed Halo-PROTAC **2** (Fig 2B: Dmax 75±5% for 1μM compound **1**, 61 ±13% for 1μM compound **3**). In contrast all three Halo-PROTACs degraded Halo-zDHHC20 (Fig 2C: Dmax 78±6% for 1μM compound **1**, 59±14% for 1μM compound **2**, 60±10% for 1μM compound **3**). We conclude from these experiments that zDHHC-PATs are amenable to PROTAC mediated degradation, and selected compound **1** for further evaluation. We evaluated off-target effects of compound **1** on endogenous zDHHC5 in Flp-In 293 T-REx cells and found no impact on its abundance (Fig 2D).

### Mechanistic characterisation of Halo-zDHHC5 and Halo-zDHHC20 degradation in Flp-In 293 T-REx cells

We next set out to confirm that the degradation of Halo-zDHHC5 and Halo-zDHHC20 induced by compound **1** was dependent on recruitment of the E3 ubiquitin ligase VHL, elongation of ubiquitin chains, and proteasome activity. Neither Halo-zDHHC5 nor Halo-

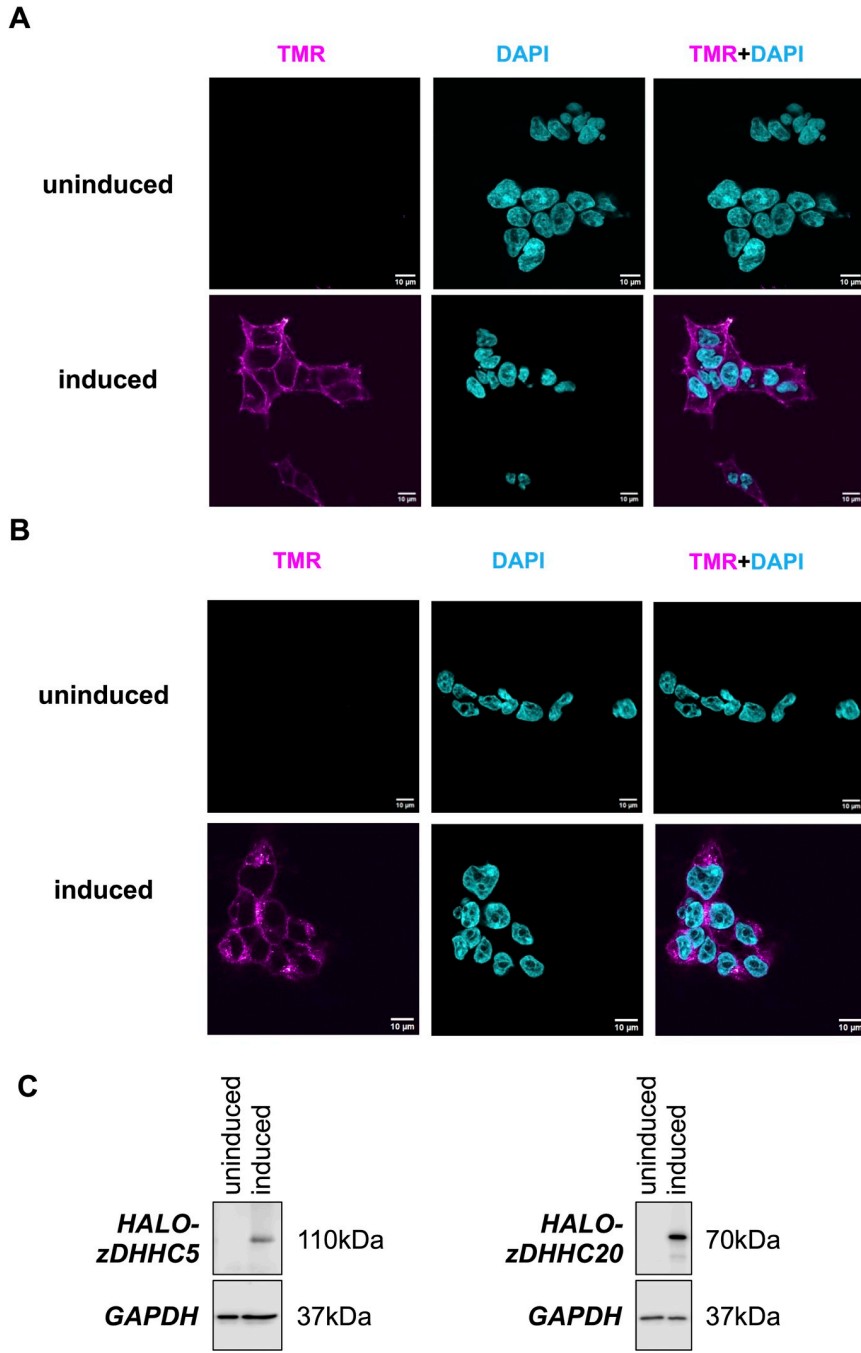

**Fig 1. Subcellular location and expression of Halo-zDHHC20 and Halo-zDHHC5 in Flp-In 293 T-REx cells.** A–Subcellular location of Halo-zDHHC5 visualised by staining cells with TAMRA-chloroalkane either without (uninduced) or with (induced) gene induction using tetracycline. B–Subcellular location of Halo-zDHHC20 visualised by staining cells with TAMRA-chloroalkane either without (uninduced) or with (induced) gene induction using tetracycline. Scale bar: 10μm. C–Western blot analysis confirming successful induction of Halo-zDHHC5 (left) and Halo-zDHHC20 (right).

zDHHC20 were degraded when cells were treated with an epimer of compound **1** in which the configuration of the prolyl hydroxyl group (established to be essential for engagement with VHL) was inverted (compound **4**, see supporting information for details). This confirms the

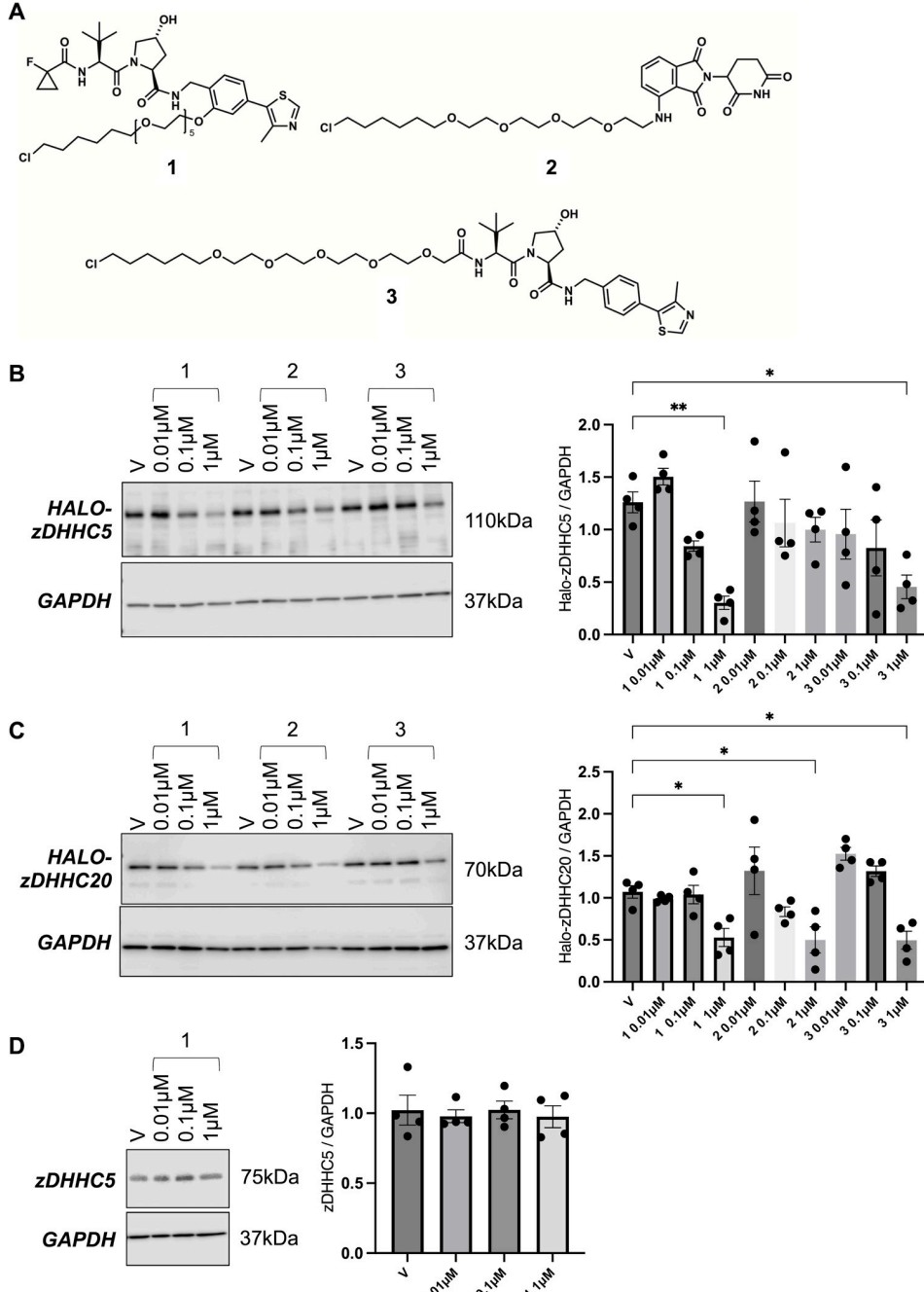

**Fig 2. Halo-PROTAC mediated zDHHC-PAT degradation in Flp-In 293 T-REx cells.** A–Halo-PROTAC structures B–Dose response relationship for Halo-PROTACs in Flp-In 293 T-REx cells expressing Halo-zDHHC5. Cells were treated with the indicated concentration of Halo-PROTAC for 18–24 hours and lysates immunoblotted as shown. V: vehicle (DMSO) control. The bar chart (right) shows the expression of Halo-zDHHC5 relative to GAPDH. *: $P < 0.05$, **: $P < 0.01$, one-way ANOVA followed by Dunnett's multiple comparisons test. C–Dose response relationship for Halo-PROTACs in Flp-In 293 T-REx cells expressing Halo-zDHHC20. Cells were treated with the indicated concentration of Halo-PROTAC for 18–24 hours and lysates immunoblotted as shown. V: vehicle (DMSO) control. The bar chart (right) shows the expression of Halo-zDHHC20 relative to GAPDH. *: $P < 0.05$, one-way ANOVA followed by Dunnett's multiple comparisons test. D–Impact of compound **1** on abundance of endogenous zDHHC5 in Flp-In 293 T-REx cells.

importance of VHL binding for Halo-zDHHC-PAT degradation ([Fig 3A]). The NEDD8 activating enzyme (NAE) is an essential component of cullin-RING ubiquitin ligases and is therefore required for ubiquitination of proteins. Inhibition of NAE with MLN4924 (10 μM) prevented Halo-zDHHC-PAT degradation by compound **1**. Similarly, inhibition of the proteasome with Mg-132 (5 μM) also prevented Halo-PROTAC induced degradation of Halo-zDHHC-PATs ([Fig 3B]), but did not alter steady-state Halo-zDHHC-PAT expression in the absence of PROTAC, suggesting the proteasome does not usually control the turnover of these proteins.

We sought direct evidence for conjugation of ubiquitin chains to Halo-zDHHC20 in the presence of compound **1**. Cells were transfected with HA-ubiquitin, treated with compound **1** and Mg-132, and Halo-tagged proteins were immunoprecipitated and immunoblotted for HA. HA-ubiquitin was incorporated into numerous cellular proteins ([Fig 3C]). Little-to-no HA-ubiquitin was incorporated into zDHHC20 in the absence of Halo-PROTAC, but treatment with compound **1** resulted in polyubiquitination of Halo-zDHHC20 ([Fig 3C]). Treatment with both compound **1** and Mg-132 (5μM) significantly increased the amount of ubiquitinated Halo-zDHHC20. Collectively these experiments support the concept that degradation Halo-zDHHC20 in cells treated with compound **1** involves the recruitment of VHL, polyubiquitination and subsequent proteasomal degradation of Halo-zDHHC20.

## Impact of Halo-zDHHC5 and Halo-zDHHC20 degradation on substrate palmitoylation in Flp-In 293 T-REx cells

Having established conditions that generated robust degradation of Halo-zDHHC5, we evaluated the impact of compound **1** on PLM palmitoylation. Palmitoylated proteins were prepared using acyl-resin assisted capture (acyl-RAC) and samples immunoblotted for proteins of interest and the constitutively palmitoylated lipid raft resident protein Flotillin-2 as an assay control [40]. [Fig 4A] demonstrates a significant decrease in PLM palmitoylation upon degradation of zDHHC5 (by 47±22% compared to vehicle treated cells). We conclude from these experiments that targeted degradation of zDHHC5 is a viable strategy to decrease palmitoylation of its substrate PLM.

To evaluate the impact of zDHHC20 degradation on substrate palmitoylation we first assessed the palmitoylation status of IFITM3, a zDHHC20 substrate which is endogenously expressed in Flp-In 293 T-REx cells [41]. Induction of Halo-zDHHC20 expression significantly increased IFITM3 palmitoylation compared to cells in which Halo-zDHHC20 was not induced ([Fig 4B]). Treatment with compound **1** significantly decreased IFITM3 palmitoylation (by 50±12% compared to vehicle treated cells expressing Halo-zDHHC20).

The SARS-CoV-2 spike protein is palmitoylated by zDHHC20 at a cluster of 10 cytosolic cysteines just proximal to its integral membrane domain [19–21]. We transfected HA-tagged SARS-CoV-2 spike into Halo-zDHHC20 expressing cells and evaluated the impact of degrading Halo-zDHHC20 on spike protein palmitoylation ([Fig 4C]). The ~210kDa spike protein is cleaved into multiple fragments by host proteases; we focussed on palmitoylation of an ~80kDa cleavage product. Neither induction of Halo-zDHHC20 expression nor Halo-zDHHC20 degradation altered SARS-CoV-2 spike protein palmitoylation in this experimental model.

## Targeting Halo-zDHHC20 in Vero E6 zDHHC20 knockout cells

We reasoned that the presence of endogenous zDHHC20 in Flp-In 293 T-REx cells, which would not be amenable to Halo-PROTAC degradation, may limit our ability to target SARS-CoV-2 spike protein palmitoylation by degrading Halo-zDHHC20. We therefore generated a cell line inducibly expressing Halo-zDHHC20 on a background of endogenous zDHHC20

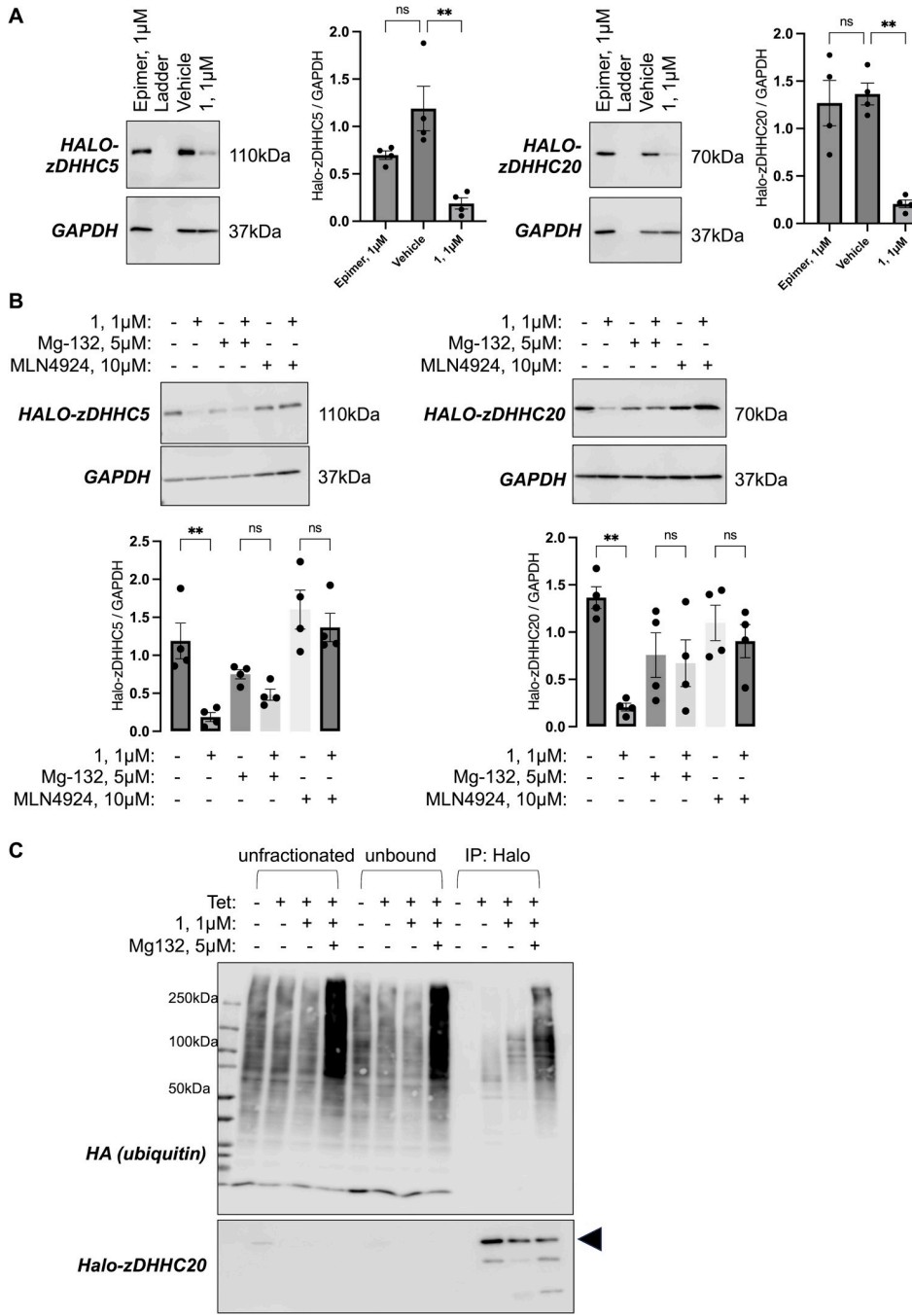

**Fig 3. Halo-PROTAC mediated zDHHC-PAT degradation requires recruitment of VHL, elongation of ubiquitin chains, and proteasome activity.** A–An epimer of compound **1** which does not recruit VHL does not induce degradation of Halo-zDHHC5 (left) or Halo-zDHHC20 (right). The bar charts show the expression of each Halo-zDHHC-PAT relative to GAPDH. **: P<0.01, one-way ANOVA followed by Tukey's multiple comparisons test. B–Inhibition of the protease using Mg-132 (5μM) or inhibition of NEDD8 activating enzyme using MLN4924 (10μM) prevent degradation of Halo-zDHHC5 (left) or Halo-zDHHC20 (right) by compound **1**. **: P<0.01, one-way ANOVA followed by Tukey's multiple comparisons test. C–Immunoprecipitation experiments confirm the incorporation of ubiquitin chains into Halo-zDHHC20 induced by treatment with compound **1**. Flp-In 293 T-REx cells in which zDHHC20 expression was induced or not with tetracycline (± Tet) were transfected with HA-ubiquitin and treated with compound (1μM) alone or in combination with Mg-132 (5μM). Whole cell lysates (unfractionated), immunoprecipitation fractions that did not bind the anti-Halo beads (unbound) and proteins immunoprecipitated by the anti-Halo beads were immunoblotted as shown.

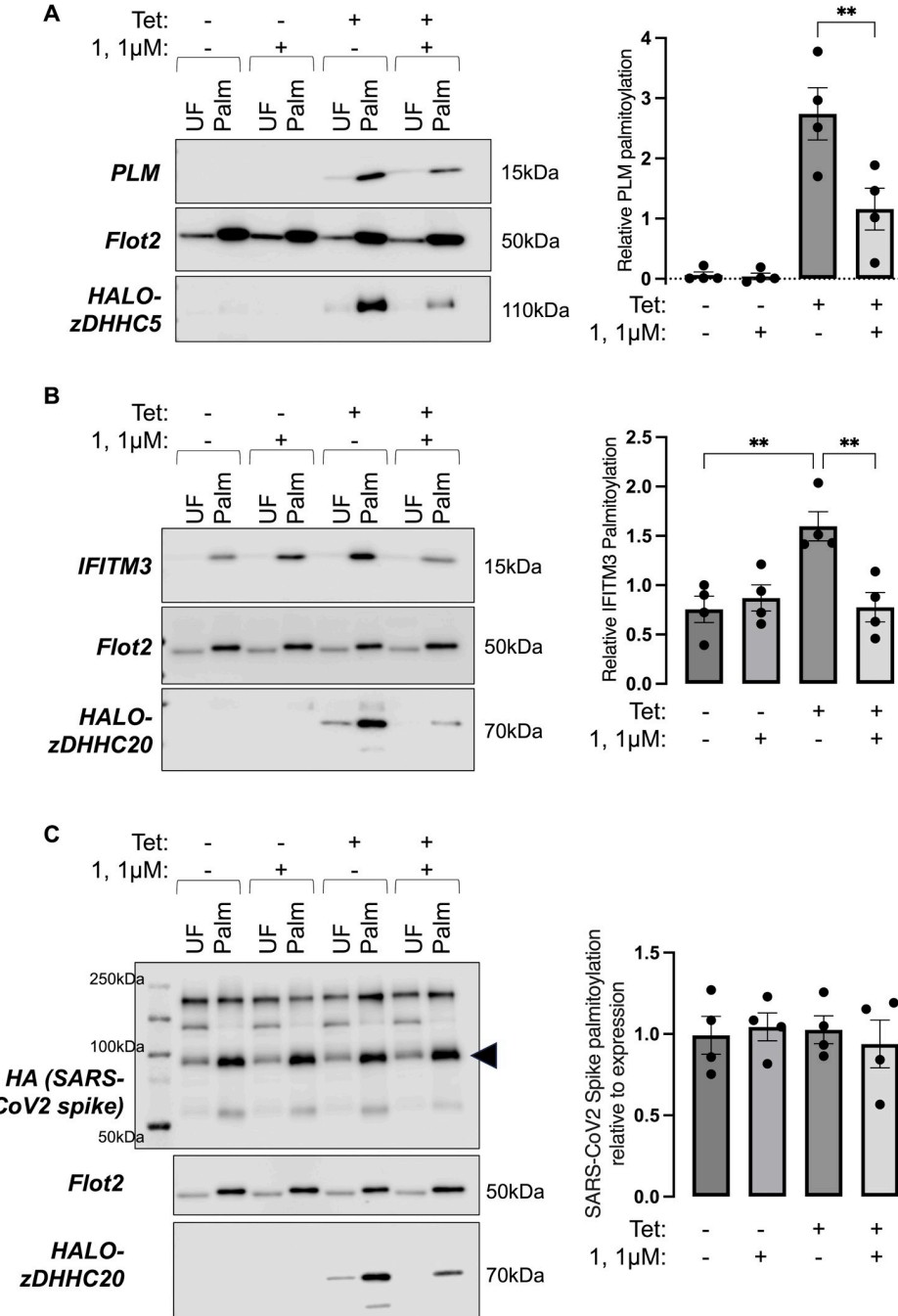

**Fig 4. Halo-PROTAC mediated zDHHC-PAT degradation decreases substrate palmitoylation in Flp-In 293 T-REx cells.** A–Palmitoylated proteins were purified from Flp-In 293 T-REx cells engineered to express tetracycline (Tet) inducible Halo-zDHHC5 and phospholemman (PLM) after expression was induced or not with tetracycline (± Tet). The Western blots show palmitoylated proteins (Palm) immunoblotted alongside corresponding unfractionated cell lysates (UF). The bar chart (right) shows the relative palmitoylation of PLM. **: P<0.01, one-way ANOVA followed by Tukey's multiple comparisons test. B–Palmitoylated proteins were purified from Flp-In 293 T-REx cells engineered to express tetracycline (Tet) inducible Halo-zDHHC20 after expression was induced or not with tetracycline (± Tet). The Western blots show palmitoylated proteins (Palm) immunoblotted alongside corresponding unfractionated cell lysates (UF). The bar chart (right) shows the relative palmitoylation of IFITM3. **: P<0.01, one-way ANOVA followed by Tukey's multiple comparisons test. C–Palmitoylated proteins were purified from Flp-In 293 T-REx cells engineered to express tetracycline (Tet) inducible Halo-zDHHC20 and transfected with HA-tagged SARS-CoV2 spike. The Western blots show palmitoylated proteins (Palm) immunoblotted alongside corresponding unfractionated cell lysates (UF). The bar chart (right) shows the palmitoylation of the SARS-CoV2 spike 80kDa cleavage product relative to its abundance in the corresponding unfractionated cell lysate.

knockout. We selected Vero E6 cells for these experiments since these cells are validated for SARS-CoV-2 replication assays and re-engineered a validated zDHHC20 knockout cell line [19] to express Halo-zDHHC20.

Halo-zDHHC20 was predominantly localised in intracellular compartments in Vero E6 cells (Fig 5A). We achieved robust expression of Halo-zDHHC20 in this model, but we were unable to degrade Halo-zDHHC20 with any Halo-PROTACs (Fig 5B). Given their efficacy in FT-293 cells, we reasoned that the presence of a drug efflux mechanism in Vero E6 cells may limit PROTAC efficacy. Co-application of verapamil (40 μM), a multi-drug resistance transporter inhibitor [42], enhanced compound **1** efficacy compared to vehicle treated cells (Fig 5C) and we therefore co-applied compound **1** with verapamil in subsequent experiments.

In zDHHC20 knockout Vero E6 cells engineered to express tetracycline inducible Halo-zDHHC20, induction of Halo-zDHHC20 expression did not increase palmitoylation of either endogenous IFITM3 or HA tagged SARS-CoV-2 spike protein expressed via transient transfection (Fig 5D). In keeping with this result, knockdown of Halo-zDHHC20 using compound **1** co-applied with verapamil did not reduce palmitoylation of either target (Fig 5D).

## Comparison of PROTAC and nanobody-mediated zDHHC20 degradation in Flp-In 293 T-REx cells

PROTAC-mediated degradation of a protein of interest is dependent on the recruitment of an E3 ubiquitin ligase. In circumstances where target abundance significantly exceeds E3 ligase abundance, or the synthesis rate of the target protein significantly exceeds the ubiquitination capacity of the ligase, PROTACs mediated knockdown may be limited. Genetically encoded nanobody-E3 ligase conjugates offer an alternative approach to target a protein of interest (for example [43]). We investigated whether a nanobody-E3 ligase conjugate offered superior degradation of zDHHC20 to a PROTAC. We engineered FT-293 cells to express tetracycline inducible Halo-zDHHC20-YFP and compared degradation induced by compound **1** (1μM) with degradation induced by transfecting cells with a fusion protein composed of an anti-GFP nanobody (LaG-16 [44]) and the HECT domain of the E3 ubiquitin ligase NEDD4L (Fig 6). PROTAC mediated knockdown was superior to nanobody mediated knockdown, and combining a Halo-directed PROTAC with a GFP-directed nanobody fused to an E3 ligase did not induce greater degradation of zDHHC20 than PROTAC alone.

## Discussion

In this investigation, we set out to evaluate the tractability of integral membrane zDHHC-PAT enzymes to PROTAC mediated degradation, and whether degrading this family of enzymes was a viable approach to target palmitoylation of their substrates. We report that zDHHC-PATs that are localised to both intracellular compartments and the cell surface membrane can be successfully degraded using PROTACs. In some settings, zDHHC-PAT degradation leads to decreased substrate palmitoylation.

The successful targeting of PLM palmitoylation by zDHHC5 in this investigation is likely a result of the relatively small number of zDHHC-PAT enzymes present at the cell surface membrane, which means that another enzyme cannot substitute when zDHHC5 is degraded. This contrasts with intracellular targets, where there is established promiscuity between enzyme / substrate pairs. For example, zDHHCs 3 and 7 represent a 'high capacity, low specificity' palmitoylation system within the secretory pathway [45], which may be a factor that limits any strategy targeting palmitoylation of particular substrates. Along similar lines, although zDHHC20 activity is a major determinant of the palmitoylation of the SARS-CoV-2 spike protein, other enzymes are also capable of palmitoylating this substrate [19, 21], which likely

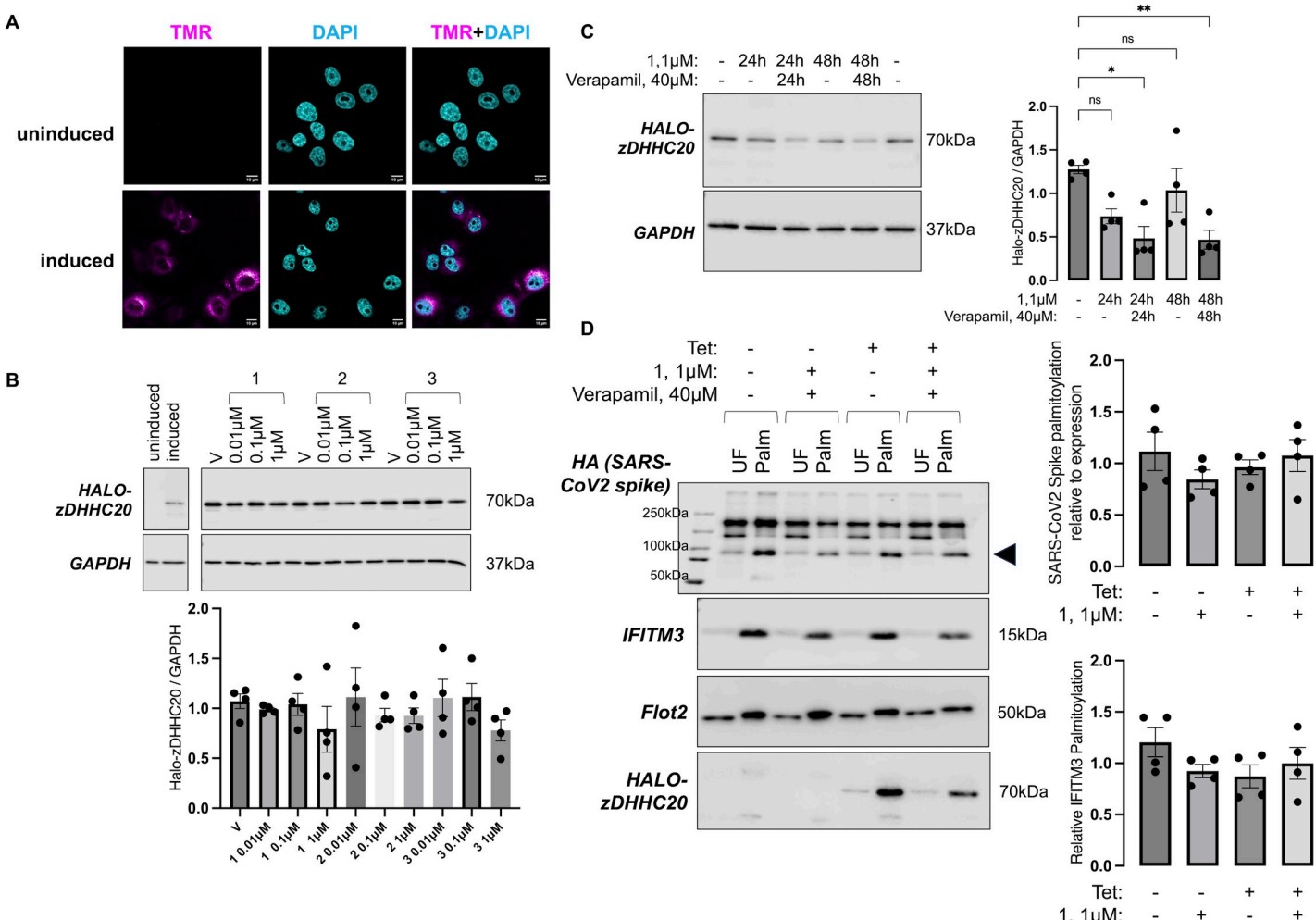

**Fig 5. Targeting Halo-zDHHC20 in Vero E6 zDHHC20 knockout cells.** A–Subcellular location of Halo-zDHHC20 visualised by staining cells with TAMRA-chloroalkane either without (uninduced) or with (induced) gene induction using tetracycline. Scale bar: 10μm. B–Western blotting confirms successful induction of Halo-zDHHC20 expression (left). Dose response relationship for Halo-PROTACs in Vero E6 zDHHC20 knockout cells expressing Halo-zDHHC20 (right). Cells were treated with the indicated concentration of Halo-PROTAC for 18–24 hours and lysates immunoblotted as shown. V: vehicle (DMSO) control. The bar chart (right) shows the expression of Halo-zDHHC20 relative to GAPDH. C–Impact of co-applying the multi-drug resistance transporter inhibitor verapamil (40μM) with compound **1** (1μM) for either 24h or 48h on Halo-zDHHC20 abundance in Vero E6 zDHHC20 knockout cells. The bar chart (right) shows the expression of Halo-zDHHC20 relative to GAPDH. *: $P<0.05$, **: $P<0.01$, one-way ANOVA followed by Tukey's multiple comparisons test. D–Palmitoylated proteins were purified from Vero E6 cells engineered to express tetracycline (Tet) inducible Halo-zDHHC20, transfected with HA-tagged SARS-CoV2 spike. The Western blots show palmitoylated proteins (Palm) immunoblotted alongside corresponding unfractionated cell lysates (UF). The bar charts show the relative palmitoylation of the SARS-CoV2 spike 80kDa cleavage product and IFITM3.

accounts for our failure to reduce spike protein palmitoylation when degrading Halo-zDHHC20. In contrast, our results indicate that while zDHHC20 abundance is a determinant of IFITM3 palmitoylation status in Flp-In 293 T-REx cells (accounting for us successfully targeting IFITM3 palmitoylation by degrading Halo-zDHHC20), zDHHC20 abundance is not rate limiting for IFITM3 palmitoylation in zDHHC20 knockout Vero E6 cells. Evidently, zDHHC-PATs expressed in Vero E6 cells can fully palmitoylate IFITM3 even in the absence of zDHHC20. Multiple zDHHC-PATs are capable of palmitoylating IFITM3 [41, 46]. The recent description of cell-specific expression maps of palmitoylating and depalmitoylating enzymes will clearly be an important tool to identify appropriate cellular models to evaluate the efficacy of zDHHC-PAT inhibitors and degraders [47].

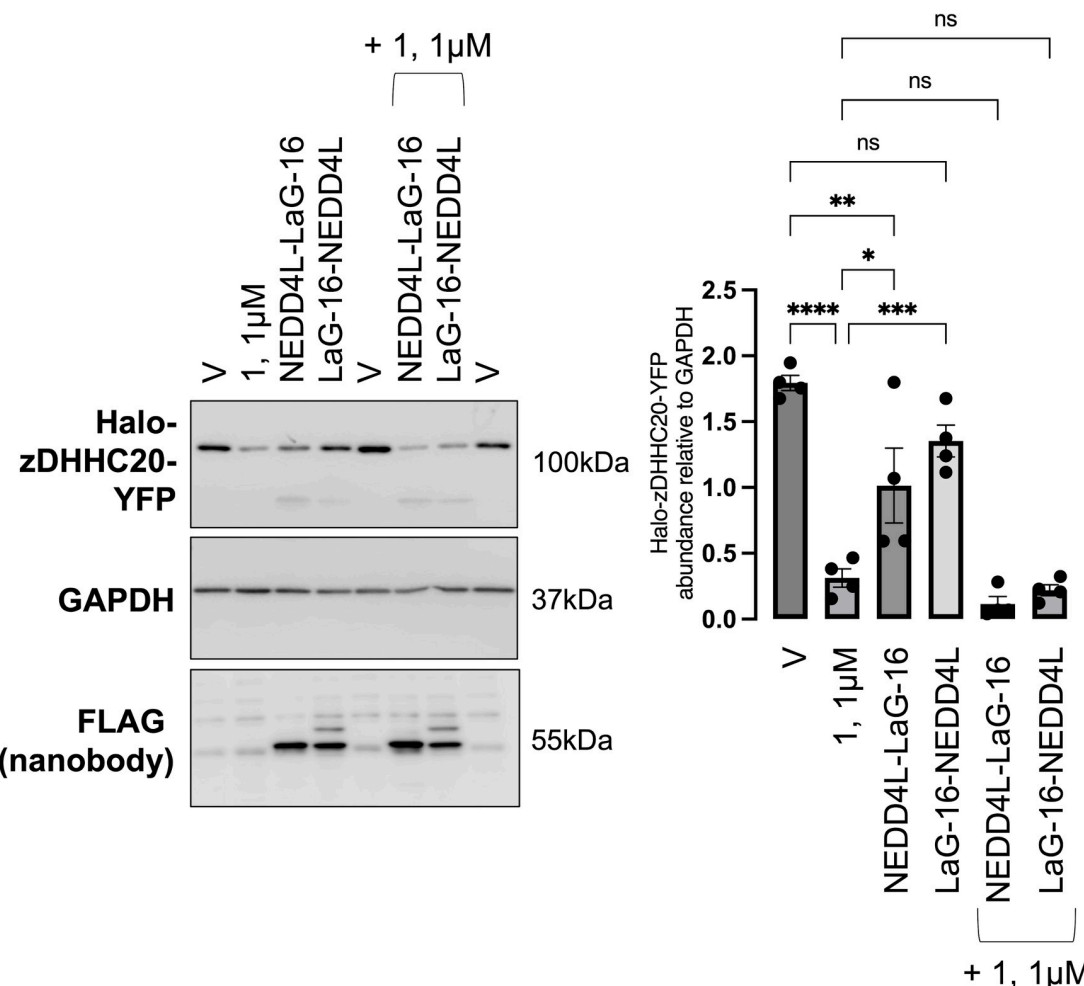

**Fig 6. Comparison of nanobody and PROTAC mediated Halo-zDHHC20-YFP degradation in Flp-In 293 T-REx cells.** Flp-In 293 T-REx cells engineered to express Halo-zDHHC20-YFP were treated with compound **1** (1μM) and / or transfected with the YFP-directed nanobody LAG-16 with the HECT Domain of NEDD4L fused at either its amino or carboxyl terminus. The bar chart (right) shows the expression of Halo-zDHHC20-YFP relative to GAPDH. *: P<0.05, **: P<0.01, ***: P<0.001, ****: P<0.0001, one-way ANOVA followed by Tukey's multiple comparisons test.

The very different subcellular localisation of zDHHC20 in Flp-In 293 T-Rex and Vero E6 cells may also account for the different dependence of IFITM3 on zDHHC20 for palmitoylation in these models. In Flp-In 293 T-Rex cells zDHHC20 localises to both intracellular membranes and the cell surface, whereas in Vero E6 cells it is predominantly intracellular. It is noteworthy that a recently-described chemical genetics system to identify zDHHC20 substrates identified numerous substrates that were only palmitoylated by zDHHC20 in one out of three cell lines investigated [48]. The different subcellular localization of zDHHC20 in different cell types may account for this cell-specific substrate preference. We are aware of no studies that have systematically evaluated subcellular localization of zDHHC-PATs in multiple cell types. One investigation investigated zDHHC-PAT subcellular distribution solely in HEK 293T cells [49]. The clear difference between zDHHC20 distribution in two cell types of similar origin identified in our investigation highlights the importance of evaluating the whole enzyme family's cellular location in more cell and tissue types in future investigations.

All PROTACs we evaluated in this study successfully degraded Halo-zDHHC20 in HEK-derived Flp-In 293 T-REx cells, but only VHL-directed PROTACs **1** and **3** were effective against Halo-zDHHC5. Clearly then, cereblon-directed PROTACs such as compound **2** can target integral membrane proteins for degradation in this cell type. The failure to target Halo-zDHHC5 with compound **2** may, be related to suboptimal geometry between the target protein and the E3 ligase in the ternary complex (**2** has a shorter linker between Halo and E3 ligase ligands than compounds **1** and **3**). Alternatively, it may be related to the subcellular location of the target protein. The ubiquitin proteasome system is particularly active early in the secretory pathway, where it functions in quality control of newly translated proteins [50, 51]. Intracellular integral membrane proteins such as a zDHHC20 may therefore be more amenable to targeted degradation than their plasma membrane equivalents. That said, both cereblon and VHL directed ligands have successfully been used to target plasma membrane resident proteins [52–55].

In conclusion, we demonstrate the tractability of zDHHC-PATs to PROTAC-mediated degradation. This significantly increases the intracellular surface area of this enzyme family that could be targeted with small molecules to achieve isoform-specific inhibition. Investigations to identify isoform-specific ligands binding outside the conserved active site of this enzyme family are now warranted.

## Methods

### Drugs

Verapamil and Mg-132 were obtained from Merck, MLN4924 was obtained from Cambridge Bioscience. All drugs were dissolved in DMSO and applied to cells from a 1000-fold concentrated stock solution. Vehicle treated cells received DMSO only.

### Antibodies

This investigation used antibodies raised to GAPDH (Merck clone GAPDH-71-1, 1:10000), Halo (Promega G9281, 1:1000–1:5000), GFP (ProteinTech clone 3H9, 1:1000), IFITM3 (Proteintech 11714-1-AP, 1:5000), PLM (Abcam ab76597, 1:1000), Flotillin 2 (BD Biosciences 610384, 1:2000), HA tag (Merck 11867423001, 1:5000), zDHHC5 (Merck, HPA014670, 1:1000).

### Plasmids and molecular biology

Plasmids encoding murine zDHHC-PATs were generously provided by Professor Masaki Fukata, National Institute for Physiological Sciences, Japan. The Halo tag cDNA was from Promega. Plasmids pcDNA5-FRT/TO and pLKO encoding Halo-zDHHC-PATs were generated by InFusion cloning (Takara) of PCR products amplified using primers designed using the Takara online primer design tool.

### Cell culture and generation of stably expressing cell lines

Flp-In 293 T-REx cells were obtained from Thermo and cultured in DMEM supplemented with 10% foetal bovine serum 15μg/ml blasticidin (Invivogen) and 100μg/ml zeocin (Invivogen). Flp-In 293 T-REx lines stably expressing proteins of interest were generated by cotransfecting cells with pcDNA 5 FRT/TO (encoding the gene of interest) and pOG44 using GeneJuice (Merck) according to the manufacturer's instructions. Stably transfected cells were selected using 100μg/ml hygromycin (Invivogen).

Lentiviruses encoding Halo-zDHHC20 were generated by transfecting 293T cells with pLKO-Halo-zDHHC20. zDHHC20 knockout Vero E6 cells (described in [19]) generously

provided by Professor Gisou van der Goot (EPFL, Switzerland) were transduced with lentiviruses and selected with 2mg/ml G418 (Thermo).

Expression of the genes of interest was achieved by treating cells with tetracycline (10μg/ml) at the time they were seeded into plates.

## Synthesis of Halo-PROTACs

Halo-PROTACs **1**–**3** and the hydroxyproline epimer control of **1** were prepared according to standard protocols as described in Supporting Information (S1 File). All final compounds were purified by reverse phase HPLC and their purity assessed by NMR (S2 File).

## Preparation of cells lysates and western blotting

Cells were seeded in 12-well plates and treated with drugs (Halo-PROTACs, Mg-132, verapamil, MLN4924) 24 hours after seeding. All drugs (or vehicle) were applied to cells for 18–24 hours. After treatment, cells were detached using a cell scraper and lysed with lysis buffer (1% Triton X-100, 0.1% SDS, and 0.1% protease inhibitor cocktail (Merck 535140) in PBS) for 30min at 4˚C. Lysates were centrifuged at 17,500g for 5min at 4˚C, insoluble material was discarded, and soluble fractions analysed using SDS PAGE and western blotting. After electrophoresis on 6–20% polyacrylamide gradient gels, proteins were transferred to PVDF membranes, blocked with 5% non-fat milk in PBS-T for one hour and incubated with the primary antibody overnight. Secondary antibodies from Jackson Immunoresearch were applied for 1 hour at room temperature, and membranes extensively washed before protein bands were visualized using enhanced chemiluminescence using a LiCOR Odyssey FC.

## Confocal microscopy

TAMRA chloroalkane (Promega, 2μM) was applied to cells in standard culture media on glass coverslips for 15min at 37˚C. Cells were then washed with PBS, incubated in standard culture media for 30min at 37˚C, then fixed and mounted on glass slides in mounting media supplemented with DAPI. Images were acquired using a Zeiss LSM880 confocal microscope with excitation and emission filters set to 543nm and 599nm respectively for TAMRA and 405nm and 459nm for DAPI.

## Immunoprecipitation

Cells were seeded in 6-well plates and treated as required. After treatment, cells were lysed for 30min using PBS supplemented with 1% Triton X-100, 0.1% SDS, 0.5 mM EDTA, and 0.1% protease inhibitor cocktail (Merck 535140). Lysates were centrifuged at 17,500g for 5min at 4˚C and insoluble material discarded. Halo tagged proteins were immunoprecipitated from the solubilised cell lysate using Halo-Trap magnetic agarose (ProteinTech) and beads washed extensively with PBS supplemented with 1% Triton X-100, 0.5 mM EDTA, and 0.1% protease inhibitor cocktail.

## Palmitoylation assays

Palmitoylated proteins were purified using resin-assisted capture (acyl-RAC) [56]. Cells were lysed in 2.5% SDS, 1% MMTS, 1mM EDTA, 100mM HEPES, pH7.5. Lysates were agitated for 4 hours at 40˚C during which time MMTS alkylated free cysteines in proteins. Excess unreacted MMTS was removed by precipitating proteins using 3 volumes of cold acetone, incubating samples at -20˚C for 20min, and recovering protein pellets by centrifuging at 17,500g for 5min. Protein pellets were extensively washed with 70% acetone, dried, and

resolubilised in 1% SDS, 1mM EDTA, 100mM HEPES, pH7.5. A sample representing the whole cell lysate ('unfractionated') was taken at this point. Palmitoylated proteins were captured by agitating resolubilised proteins for 2.5 hours with thiopropyl Sepharose (Cytiva) in the presence of 250mM neutral hydroxylamine to cleave thioester bonds and reveal previously palmitoylated cysteines. Following capture of palmitoylated proteins, Sepharose resin was washed extensively with 1% SDS, 1mM EDTA, 100mM HEPES, pH7.5 and proteins eluted using Laemmli buffer supplemented with 100mM DTT.

## Statistical analysis

Uncropped Western blot and gel images are provided in S1 Raw images. All Western blot data were quantified using Image Studio Lite Ver 5.2. Protein abundance in whole cell lysates was normalised to the abundance of GAPDH. For genomically-expressed palmitoylated proteins (PLM, IFITM3) the amount captured in the acyl-RAC assay was normalised to amount of the palmitoylated housekeeping protein Flotillin 2 captured. For transfected proteins (e.g. SARS-CoV-2 spike protein, where expression sometimes varied from well to well) the amount captured in the acyl-RAC assay was normalised to expression. Expression / palmitoylation values for individual samples within an experiment were normalised to the mean value for all samples in that experiment. Replicate experiments were analysed using GraphPad Prism. P values were calculated using one-way ANOVA followed by appropriate post-hoc tests.

## Supporting information

**S1 Raw images. Uncropped blot & gel images.**
(PDF)

**S1 File. General chemistry experimental details.** Synthesis and evaluation of Halo-PROTACs.
(DOCX)

**S2 File. NMR analyses of compounds 1–4 used in this investigation.**
(DOCX)

## Author Contributions

**Conceptualization:** David J. France, William Fuller.

**Funding acquisition:** Michael J. Shattock, David J. France, William Fuller.

**Investigation:** Mingjie Bai, Emily Gallen, Sarah Memarzadeh, Jacqueline Howie, Xing Gao, Chien-Wen S. Kuo.

**Methodology:** Elaine Brown, Simon Swingler.

**Supervision:** Sam J. Wilson, Michael J. Shattock, David J. France, William Fuller.

**Writing – original draft:** Sarah Memarzadeh, David J. France, William Fuller.

**Writing – review & editing:** Sarah Memarzadeh, Simon Swingler, Sam J. Wilson, Michael J. Shattock, David J. France, William Fuller.

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
