## [Decision Letter · Decision Letter 0]

26 Nov 2023

PONE-D-23-35294Targeted degradation of zDHHC-PATs decreases substrate S-palmitoylationPLOS ONE

Dear Dr. Fuller,

Thank you for submitting your manuscript to PLOS ONE. After careful consideration, we feel that it has merit but does not fully meet PLOS ONE’s publication criteria as it currently stands. Therefore, we invite you to submit a revised version of the manuscript that addresses the points raised during the review process.

We look forward to receiving your revised manuscript.

Kind regards,

Jin-Ming Zhou

Academic Editor

PLOS ONE

Journal Requirements:

   "This work was supported by the British Heart Foundation, grant RG/17/15/33106 to MJS and WF and a Daphne Jackson Fellowship funded by Medical Research Scotland awarded to SS"

   "This work was supported by the British Heart Foundation, grant RG/17/15/33106 "

   "This work was supported by the British Heart Foundation, grant RG/17/15/33106 "

Reviewers' comments:

Reviewer's Responses to Questions

**Comments to the Author**

1. Is the manuscript technically sound, and do the data support the conclusions?

Reviewer #1: Yes

Reviewer #2: Yes

2. Has the statistical analysis been performed appropriately and rigorously? 

Reviewer #1: Yes

Reviewer #2: Yes

3. Have the authors made all data underlying the findings in their manuscript fully available?

Reviewer #1: Yes

Reviewer #2: Yes

4. Is the manuscript presented in an intelligible fashion and written in standard English?

Reviewer #1: Yes

Reviewer #2: Yes

5. Review Comments to the Author

Reviewer #1: This manuscript presents data on a possible method to evaluate the integral membrane zDHHC-PAT enzymes to PROTAC mediated degradation. The authors report that zDHHC-PATs that are localized to both intracellular compartments and the cell surface membrane can be successfully degraded using PROTACs, thereby leading to decrease of substrate palmitoylation. Nonetheless there are a number of issues to be addressed:

1. P7, line 161~163, “by cereblon-directed Halo-PROTAC 2 (Figure 2b: Dmax 75±5% for compound 1, 61±13% for compound 3). In contrast all three Halo-PROTACs degraded Halo-zDHHC20 (Figure 2c: Dmax 78±6% for compound 1, 59±14% for compound 2, 60±10% for compound 3).” Please show the concentration of these Halo-PROTACs in the text.

2. Figure 2C, this assay testified that degradation Halo-zDHHC20 in cells treated with compound 1 involves the polyubiquitination and subsequent proteasomal degradation of Halo-zDHHC20. One way to make the result more robust would be to set a control which add Tet and MG132 but without PROTAC compound 1; this might enable observation of a more convincing result.

3. Figure 4. The reason for using Flot2 as an internal control should be mentioned and explained in the text.

4. Vero E6 zDHHC20 knockout cell line should be verified by immunoblotting or sequencing.

5. P10, line 254~255, “but we were unable to degrade Halo-zDHHC20 with any Halo-PROTACs (Figure 5b).” However, in figure 5B, it seems that compound 1 significantly decrease the Halo-zDHHC20 level at the concentration of 1µM in WB result.

6. In Figure 5C, the results shown in the chart are not very consistent with those in WB. Obviously, in WB result, the verapamil enhanced compound 1 efficacy on Halo-zDHHC20 degradation more potent at 24 hours than that at 48 hours.

Reviewer #2: This is a review of the paper “Targeted degradation of zDHHC-PATs decreases substrate S-palmitoylation” for PLOS ONE.

This paper points out that the inhibition of the activity of individual zDHHC-PATs is reasonable, but the specificity of existing tools is poor. Thus, the authors developed 3 molecules based on PROTACs to target this class of enzymes. They evaluated the impact of Halo-PROTACs on zDHHC-PAT expression and substrate palmitoylation. While, they concluded that although PROTAC-mediated targeting of zDHHC-PATs to decrease substrate palmitoylation is feasible, given the well-established degeneracy in the zDHHC-PAT family, in some settings the activity of non-targeted zDHHC-PATs may substitute and preserve substrate palmitoylation.

Some changes that are recommended.

1. Figure 1, the author only detected tetracycline induced protein expression by Confocal microscopy. However, before this assay, I advised they could detect the total protein and mRNA expression levels by Western blot and qPCR.

2. It is not necessary to show the protein marker in the WB pictures, the author should clip the original image and make them showed in general types, such as cropped gel band, band molecular weight, cell lines name, band’s name labeled in the left of the panel.

3. Line 250-252, the authors should provide data that proved the endogenous zDHHC20 was knockout.

4. It is recommended that introduce the Halo-PROTACs technology and its application in zDHHC-PATs degradation in the introduction chapter.

5. The author should detect the cell toxicity of these compound.

6. The writing in the Methods chapter is overly simplistic and many important details are not clearly written.

7. The author should perform additional assay to prove the specificity of compound 1.

Overall, after making corrections and additions, I think this is a useful paper to publish.

6. PLOS authors have the option to publish the peer review history of their article (what does this mean?). If published, this will include your full peer review and any attached files.

Reviewer #1: No

Reviewer #2: No

---

## [Author Response · Author response to Decision Letter 0]

1 Feb 2024

Summary of Changes

We are grateful to the reviewers and editor for their generally positive assessment of our manuscript. We particularly thank the reviewers for their suggestions to improve clarity and support our conclusions. We have undertaken new experiments and analysis to address most of the reviewers’ comments.

Figure 1: Added new panel C demonstrating successful induction of HALO-tagged zDHHC-PATs using western blotting.

Figure 2: Assessed off-target effects of compound 1 on un-tagged zDHHC-PATs by evaluating its impact on endogenous zDHHC5 (panel D).

Figure 5: Added new data to panel B confirming successful induction of HALO-zDHHC-20 using western blotting.

Editor Comments:

Confirmed

 "This work was supported by the British Heart Foundation, grant RG/17/15/33106 to MJS and WF and a Daphne Jackson Fellowship funded by Medical Research Scotland awarded to SS"

 "This work was supported by the British Heart Foundation, grant RG/17/15/33106 "

Confirmed in the cover letter.

 "This work was supported by the British Heart Foundation, grant RG/17/15/33106 "

Confirmed in the cover letter.

Confirmed.

Confirmed

Confirmed

 

Reviewer #1: This manuscript presents data on a possible method to evaluate the integral membrane zDHHC-PAT enzymes to PROTAC mediated degradation. The authors report that zDHHC-PATs that are localized to both intracellular compartments and the cell surface membrane can be successfully degraded using PROTACs, thereby leading to decrease of substrate palmitoylation. Nonetheless there are a number of issues to be addressed:

1. P7, line 161~163, “by cereblon-directed Halo-PROTAC 2 (Figure 2b: Dmax 75±5% for compound 1, 61±13% for compound 3). In contrast all three Halo-PROTACs degraded Halo-zDHHC20 (Figure 2c: Dmax 78±6% for compound 1, 59±14% for compound 2, 60±10% for compound 3).” Please show the concentration of these Halo-PROTACs in the text.

We have added the information the reviewer requested, lines 175-177.

2. Figure 2C, this assay testified that degradation Halo-zDHHC20 in cells treated with compound 1 involves the polyubiquitination and subsequent proteasomal degradation of Halo-zDHHC20. One way to make the result more robust would be to set a control which add Tet and MG132 but without PROTAC compound 1; this might enable observation of a more convincing result.

We believe the reviewer is referring to Fig 3C. The reviewer is correct that this would tell us whether the proteasome is involved in the turnover of Halo-zDHHC20 in the absence of PROTAC. We highlight that the expression data provided in Figure 3B largely rules out a role for the proteasome in HALO-zDHHC turnover (identical expression of HALO-zDHHCs ± Mg-132). We have noted this information in the results, lines 197-198. Ultimately, even if HALO-zDHHC turnover is controlled by the proteasome, we don’t think this result would be particularly impactful (it would not be relevant for the endogenous zDHHCs – the system is too artificial to make this point worth chasing for now).

3. Figure 4. The reason for using Flot2 as an internal control should be mentioned and explained in the text.

We have added the information the reviewer requested to description of Figure 4, lies 228-230.

4. Vero E6 zDHHC20 knockout cell line should be verified by immunoblotting or sequencing.

This cell line was provided by a collaborator who published work confirming the absence of zDHHC20. We have added this information to the results section, line 274.

5. P10, line 254~255, “but we were unable to degrade Halo-zDHHC20 with any Halo-PROTACs (Figure 5b).” However, in figure 5B, it seems that compound 1 significantly decrease the Halo-zDHHC20 level at the concentration of 1µM in WB result.

We apologise. On average 1µM did not have any meaningful impact on expression, but we picked a bad representative blot and have replaced it with an alternative replicate.

6. In Figure 5C, the results shown in the chart are not very consistent with those in WB. Obviously, in WB result, the verapamil enhanced compound 1 efficacy on Halo-zDHHC20 degradation more potent at 24 hours than that at 48 hours.

We apologise – we picked a bad representative blot and have replaced it with an alternative replicate.

 

Reviewer #2: This is a review of the paper “Targeted degradation of zDHHC-PATs decreases substrate S-palmitoylation” for PLOS ONE.

This paper points out that the inhibition of the activity of individual zDHHC-PATs is reasonable, but the specificity of existing tools is poor. Thus, the authors developed 3 molecules based on PROTACs to target this class of enzymes. They evaluated the impact of Halo-PROTACs on zDHHC-PAT expression and substrate palmitoylation. While, they concluded that although PROTAC-mediated targeting of zDHHC-PATs to decrease substrate palmitoylation is feasible, given the well-established degeneracy in the zDHHC-PAT family, in some settings the activity of non-targeted zDHHC-PATs may substitute and preserve substrate palmitoylation.

Some changes that are recommended.

1. Figure 1, the author only detected tetracycline induced protein expression by Confocal microscopy. However, before this assay, I advised they could detect the total protein and mRNA expression levels by Western blot and qPCR.

Thank you for the suggestion. We have added Western blots to Figure 1 and Figure 5 to demonstrate that the cell lines used express a Halo-tagged protein of the predicted molecular weight in a tetracycline-inducible manner.

2. It is not necessary to show the protein marker in the WB pictures, the author should clip the original image and make them showed in general types, such as cropped gel band, band molecular weight, cell lines name, band’s name labeled in the left of the panel.

We have cropped off all the ladders as suggested, except for a couple of examples where the ladder helps the reader understand the experiment better (ubiquitination experiment, spike protein).

3. Line 250-252, the authors should provide data that proved the endogenous zDHHC20 was knockout.

This cell line was provided by a collaborator who published work confirming the absence of zDHHC20. We have added this information to the results section, line 274.

4. It is recommended that introduce the Halo-PROTACs technology and its application in zDHHC-PATs degradation in the introduction chapter.

Thank you for the recommendation. We have added the information the reviewer suggests. See lines 117-121.

5. The author should detect the cell toxicity of these compound.

We did not formally test toxicity of the Halo-PROTACs but we observed no impact on cell growth or morphology using them up to 10µM (10x higher than the highest concentration reported in our investigation). These compounds have been widely used in the literature and others have already reported no toxicity when used below 10µM (ACS Chem Biol 10, 1831-1837 (2015)). We have added this information to the introduction, lines 117-121.

6. The writing in the Methods chapter is overly simplistic and many important details are not clearly written.

We have rewritten several sections of the methods to improve clarity.

7. The author should perform additional assay to prove the specificity of compound 1.

Thank you for the suggestion. We evaluated whether compound 1 impacts expression of endogenous zDHHC-PATs by probing for zDHHC5 in cells treated with compound 1. There is no impact – the data is presented in Figure 2d and lines 182-183. We cannot do the same for zDHHC20 because the antibodies for the endogenous protein are unreliable in our hands. The striking specificity of Halo-PROTACs at a whole proteome level has been described by others, which we now mention in the introduction (lines 117-121).

Overall, after making corrections and additions, I think this is a useful paper to publish.

---

## [Decision Letter · Decision Letter 1]

14 Feb 2024

Targeted degradation of zDHHC-PATs decreases substrate S-palmitoylation

PONE-D-23-35294R1

Dear Dr. Fuller

We’re pleased to inform you that your manuscript has been judged scientifically suitable for publication and will be formally accepted for publication once it meets all outstanding technical requirements.

Kind regards,

Jin-Ming Zhou

Academic Editor

PLOS ONE

Additional Editor Comments (optional):

Reviewers' comments:

Reviewer's Responses to Questions

**Comments to the Author**

1. If the authors have adequately addressed your comments raised in a previous round of review and you feel that this manuscript is now acceptable for publication, you may indicate that here to bypass the “Comments to the Author” section, enter your conflict of interest statement in the “Confidential to Editor” section, and submit your "Accept" recommendation.

Reviewer #2: All comments have been addressed

2. Is the manuscript technically sound, and do the data support the conclusions?

Reviewer #2: Yes

3. Has the statistical analysis been performed appropriately and rigorously? 

Reviewer #2: Yes

4. Have the authors made all data underlying the findings in their manuscript fully available?

Reviewer #2: Yes

5. Is the manuscript presented in an intelligible fashion and written in standard English?

Reviewer #2: Yes

6. Review Comments to the Author

Reviewer #2: The authors have adequately addressed my comments raised in a previous round of review and I think this manuscript is now acceptable for publication

7. PLOS authors have the option to publish the peer review history of their article (what does this mean?). If published, this will include your full peer review and any attached files.

Reviewer #2: No

---

## [Editor Report · Acceptance letter]

11 Mar 2024

PONE-D-23-35294R1 

PLOS ONE

Dear Dr. Fuller, 

I'm pleased to inform you that your manuscript has been deemed suitable for publication in PLOS ONE. Congratulations! Your manuscript is now being handed over to our production team.

Kind regards, 

on behalf of

Dr. Jin-Ming Zhou 

Academic Editor

PLOS ONE